

# A statistically based seasonal precipitation forecast model with automatic predictor selection and its application to Central and South Asian headwater catchments

Lars Gerlitz, Heiko Apel, Abror Gafurov, Katy Unger-Shayesteh, Sergiy Vorogushyn, & Bruno Merz

GFZ German Research Center for Geosciences, Section 5.4: Hydrology, Telegrafenberg, 14473 Potsdam, Germany

*Correspondence to:* Lars Gerlitz (lars.gerlitz@gfz-potsdam.de)

**Abstract.** The study presents a statistically based seasonal precipitation forecast model, which automatically identifies suitable predictors from globally gridded SST and climate variables by means of an extensive data mining procedure and explicitly avoids the utilization of typical large-scale climate indices. This leads to an enhanced flexibility of the model and enables its automatic calibration for any target area without any prior assumption concerning adequate predictor variables. Potential predictor variables are derived by means of a cellwise correlation analysis of precipitation anomalies with gridded global climate variables under consideration of varying lead times. Significantly correlated grid cells are subsequently aggregated to predictor regions by means of a variability based cluster analysis. Finally for every month and lead time, an individual random forest based forecast model is constructed, by means of the preliminary generated predictor variables. Monthly predictions are aggregated to running three-months periods in order to generate a seasonal precipitation forecast.

The model is exemplarily applied and evaluated for selected headwater catchments in Central and South Asia. Particularly for winter and spring precipitation (which is associated with a westerly flow in the entire target domain) the model shows solid results with correlation coefficients up to 0.7 between forecasted and observed precipitation, although the variability of precipitation rates is strongly underestimated. Likewise for the monsoonal precipitation amounts in the South Asian target areas a promising skill of the model could be detected. The skill of the model for the dry summer season in Central Asia and the transition seasons over South Asia is found to be low.

A sensitivity analysis by means of well-known climate indices (such as the El Nino Southern Oscillation and the North Atlantic Oscillation) reveals the major large-scale controlling mechanisms of the seasonal precipitation climate for each target area. For the Central Asian target areas, both, ENSO and NAO are identified as important controlling factors for precipitation totals during moist spring season. Drought conditions are found to be triggered by a warm ENSO phase in combination with a positive phase of the NAO. For the monsoonal summer precipitation amounts over Southern Asia, the model suggests a distinct negative response to El Nino events.

## 1   Introduction

Seasonal precipitation prediction is a crucial task in the field of applied climatology, particularly due to the manifold ecological, economic and social consequences of abnormal weather conditions, such as droughts and flood events. Especially in regions, characterized by a large inter-annual precipitation variability, a seasonal forecast hydro-climatological parameters is required by governmental and non-governmental stakeholders in order to develop and implement adequate adaption strategies e.g. for water resource management and flood protection (Chiew et al., 2003).





In general precipitation is a result of complex and interacting atmospheric phenomena at different spatial and temporal
scales and is highly variable in space and time. Thus its precise prediction more than several days ahead is illusive.
However regional climate conditions are actively involved in large-scale atmospheric patterns, which are (1) occasionally
persistent and (2) influenced by lateral boundary conditions, such as sea surface temperatures, land cover and soil moisture
and by external factors, e.g. variations of the solar radiation and volcanic eruptions (Palmer and Anderson, 1994; Smith
et al., 2012) . The fact that the boundary conditions are often characterized by a low frequency variability leads to a certain
predictabilty of medium range climate conditions in many regions of the world.
Operational seasonal forecasts are usually based on dynamical Atmosphere Ocean General Circulation Models
(AOGCMs). These process-based models enable the prediction of large-scale climate conditions and variations at various
temporal scales (Saha et al., 2014; Smith et al., 2012). Based on the fundamental fluid dynamic equations these models
are designed to simulate large-scale characteristics of the climate system in a physically consistent manner. With regard
to exponentially increasing computing demands, the equations are numerically solved on a coarse regular grid. Small
scale processes, such as convective precipitation or the turbulent transport of energy and motion are only indirectly
considered by means of empirically based parameterizations (Smith et al., 2012). In order to utilize AOGCMs for seasonal
climate forecasts, the models are forced with real time initial and boundary conditions. Especially tropical sea surface
temperatures, but also snow covered areas and soil moisture have been identified as important influencing factors for the
global circulation (Brands et al., 2012; Douville and Chauvin, 2000; van den Hurk et al., 2010; Orsolini et al., 2013). Best
results of process-based seasonal climate forecasts are usually found in the tropics, where large-scale wind fields and
associated moisture fluxes are highly influenced by sea surface temperature variations. Skill for the temperate climate
zones are mostly lower (Kumar et al., 2013). In general, dynamical climate models are prone to biases due to uncertainties
in the initial conditions and are particularly reliable, when large model ensembles are available (Eden et al., 2015; Suárez-
Moreno and Rodríguez-Fonseca, 2015). Due to their high computing requirements, dynamical seasonal forecasts are
reserved to a few research centers and are not suitable for application in hydro-meteorological and environmental offices,
particularly in developing and transition countries.
As an efficient alternative, statistical forecast models are widely applied in order to derive suitable input data for climate
impact investigations. Based on the assumption that seasonal climate anomalies are triggered by variations of nearby or
remote atmospheric, oceanic or terrestrial conditions, these models attempt to find robust statistical relationships between
observed climate anomalies and the state of adequate predictor variables during the previous months. Since near surface
temperature and precipitation are the most decisive variables for the hydrological budget and exhibit the strongest impact
on climate sensitive environments, these variables are frequently used as predictants.
Particularly the state of the El Nino Southern oscillation (ENSO) is known to influence the large-scale precipitation
patterns almost everywhere on the globe (Dai and Wigley, 2000; Mason and Goddard, 2001; Stone et al., 1996).
Particulary the precipitation variability in the tropical regions is directly determined by ENSO due to its impact on the
tropical Walker Circulation. During El Nino events positive SST anomalies occur over the eastern tropical Pacific as a
result of weakened easterly trade winds. A common consequence is the occurrence of drought periods in South East Asia,
especially over Indonesia, and the simultaneous presence of long lasting precipitation events over the arid regions of the
western slopes of the South American Andes (Julian and Chervin, 1978; Wang, 2002). However several studies
demonstrate a statistically significant correlation of El Nino-Indices (usually derived from SST-observations in the El
Nino core regions or from associated pressure gradients between Darvin and Tahiti) with seasonal precipitation time series
in other parts of the tropics and also in temperate climate zones. E.g. various studies detected a robust statistical
relationship between Australian monsoonal precipitation and the ENSO state during previous months (Cai et al., 2011;
Ummenhofer et al., 2009). A significant influence of El Nino events was found for monsoonal precipitation amounts in



Eastern and Southern Africa (Liebmann et al., 2014; Ratnam et al., 2014) and the Sahel region (Parhi et al., 2015). For
the South Asian monsoon a negative response to El Nino events has been frequently perceived (Krishnaswamy et al.,
2014; Lau and Wu, 2001; Surendran et al., 2015). For the semi-arid regions of Central Asia and for the Mediterranean
region a positive relationship of winter and spring precipitation to El Nino events during previous autumn was found e.g.
by Barlow et al. (2002); Hoell et al. (2013); Roghani et al. (2015) and Syed et al. (2006).  Moreover, Fraedrich (1994)
and Wu and Lin (2012) detected a statistically significant influence on extra tropical circulation anomalies such as the
position of large-scale rossby waves and the associated the North Atlantic Oscillation. This subsequently leads to a certain
impact of El Nino events on the European winter climate, although correlations are in general less robust compared with
tropical regions. Other tropical SST modes frequently used in seasonal forecasts include the Indian Ocean Dipole (IOD),
the Atlantic Multidecadal Oscillation (AMO) and the Pacific Decadal Oscillation (PDA), which have a significant
predictive skill for their adjacent coastal regions (Eden et al., 2015).
Numerous studies additionally use customized SST indices as predictor variables for seasonal precipitation forecasts. For
example Hartmann et al. (2016) tested the predictive skill of mean SSTs  from various ocean basins surrounding the Asian
continent for the precipitation variability in the arid Tarim basin in North Western China. Hertig and Jacobeit (2010)
investigated the predictive skill of EOF-derived SST patterns of the Northern Atlantic, in order to forecast winter
precipitation amounts in the Mediterranean. Seibert et al., (2016) recently demonstrated that customized SST indices from
the Indian and Southern Atlantic Ocean improve the quality of statistical seasonal forecasts for the Limpopo-basin in
Southern Africa. Suárez-Moreno and Rodríguez-Fonseca (2015) showed that particularly for coastal regions, adjacent
Sea surface temperatures can significantly improve the seasonal forecast of precipitation.
Few studies utilize large-scale atmospheric pressure modes for seasonal climate predictions. However, e.g. Wu et al. (2009)
report, that the winter state of the North Atlantic Oscillation (defined as the pressure gradient between the Iceland low
and the Azores high pressure cell) influences the SST pattern of the Northern Atlantic during spring season and affects
the intensitivity of the subsequent East Asian Summer Monsoon via cross Eurasian teleconnections. Hasson et al. (2014)
found a statistical significant influence of the North Atlantic Oscillation on winter precipitation amounts in the Indus
basin. Likewise Hartmann et al. (2016)  tested the predictive skill of pressure patterns over Europe and Asia (such as the
North Atlantic Oscillation or the Siberian High Index) on precipitation anomalies in the Tarim Basin.
Eventually local land cover characteristics are frequently applied in statistical seasonal forecast models. For example
Cohen and Entekhabi (1999) and Cohen and Barlow (2005) show that the snow cover over Eurasia during autumn and
spring alters the large-scale atmospheric circulation over the Northern hemisphere with wide implications on precipitation
patterns during subsequent months. Brands et al. (2012) report a statistical significant relationship between late autumn
snow cover over Eurasia and winter time precipitation over Europe. Tian and Fan (2015) argue, that the state of the NAO
and the associated precipitation patterns over Europe are influenced by both, Atlantic SSTs and snow cover rates over
Eurasia. Likewise some studies indicate a negative response of the South Asian monsoon to higher snow cover rates over
Eurasia, most likely due to a delayed surface heating of the Asian continent (Wu and Qian, 2003; Zhang et al., 2004).
Recently some studies also included local soil moisture or previous rainfall into statistical forecasting models in order to
capture water recycling due to authochthonous weather conditions and persistent circulation characteristics (Eden et al.,
2015; van den Hurk et al., 2010).
As shown, most statistical forecast applications utilize either well-known climate indices or expert knowledge based
customized indices from SSTs or land cover characteristics. Customized indices are frequently included, since typical
climate indices do not cover regional scale anomalies of SST or pressure patterns which might be important predictor
variables for seasonal climate forecasts in certain regions. However these customized indices are usually calibrated with





regard to specific target areas and thus are not transferable to any other regions. Hence state of the art seasonal climate
forecast models are either based on a fixed number of climate indices (and thus might not consider important predictor
variables) or are highly site specific and barely transferable to other regions. Recently some advances towards an
automatic predictor selection were made by Suárez-Moreno and Rodríguez-Fonseca (2015), who used gridded SST fields
as potential predictors in order to automatically identify SST patterns, which are relevant for the seasonal precipitation
forecast in selected target areas.
With the aim of developing an operational seasonal forecast model, which is easily transferable to any region in the world,
we present a generic data mining approach which automatically selects potential predictors from gridded SST
observations and large-scale atmospheric circulation patterns derived from reanalysis data. Subsequently the approach
generates robust statistical relationships with posterior precipitation anomalies for user selected target regions. Therefore
the statistical package R (R Development Core Team, 2008) as well as the scripting environment of the free and open
source GIS system SAGA (Conrad et al., 2015) are utilized. The precipitation forecast model is based on a cell-wise
correlation analysis of various gridded variables with regional precipitation estimates, which identifies grid cells with
potential predictive skill for a specific target area with different time lags. Grid cells which significantly correlate with
precipitation anomalies during subsequent months are aggregated to predictor regions by means of an automatic cluster
analysis for every variable and time lag. Thus, for every target area, specific predictor variables are automatically derived.
The cluster regions are afterwards utilized as potential predictors in a non-parametric and non-linear random forest based
modelling approach. Based on an independent period, the model performance for the selected target area is evaluated
before an operational forecast is generated based on real time predictor fields.
In the following section we provide a detailed overview about the utilized data sets and the main model components, such
as predictor selection, model calibration and evaluation. Subsequently we exemplary provide some applications of the
model for selected target areas in Central and South Asia. In order to make the individual modelling steps more
comprehensible, we already provide some major interim results for one target area within the following section. The target
area of the Naryn headwater catchment is located in Kyrgyzstan and is mostly influenced by westerly air masses.
Precipitation observations show a clear annual cycle with maximum precipitation amounts during spring season and a
large inter-annual variability due to its location in continental Central Asia. A detailed description of the climatic
characteristics of the Naryn basin is provided in Sec. 3.
**2    Methods and Data**
**2.1 Modelling Structure**
The major objective of the presented model is to derive suitable predictor variables from global oceanic and atmospheric
fields and to develop robust statistical relationships which enable a seasonal precipitation forecast for user selectable
target regions. The underlying data sets as well as the major model components are summarized in Fig. 1. In order to
analyze the precipitation variability in selected target areas the model is based on the CRU TS 2.0 precipitation data set,
which provides monthly precipitation estimates for the 20th century on a global grid with a resolution of 0.5° lat./long.
(Harris et al., 2014; New et al., 1999). The data set is based on a dense network of observations for the period from 1961
to 1990, which were used for the spatialization of monthly mean precipitation amounts, and a compilation of station
records with longer time series available, which were used for the calculation of anomalies and were subsequently
spatially interpolated based on inverse distances. New et al. (1999) show that this approach is suitable for the resolution



of 0.5° since it combines a climatic baseline, which is highly influenced by the underlying topography with simple
interpolated anomalies, which are mainly driven by large-scale weather conditions.
[Fig. 1]
Areal mean monthly precipitation sums for the selected target region are extracted from the CRU data set. Since monthly
time series of precipitation are usually positively skewed, which might not compromise the assumptions of the subsequent
correlation analysis, the actual values are converted into the Standardized Precipitation Index (SPI) (Guttman, 1998;
McKee et al., 1993) for every single month of the year. Therefore the precipitation distribution for each month is fitted to
a gamma distribution with suitable shape and scale parameters. The exceedance probability of observed precipitation
amounts is then converted into z-values of the normal distribution. The SPI values, which are normally distributed by
definition, are subsequently cell-wise correlated with gridded global SST and climate data with lead times ranging from
1 to 6 months. For every variable and lead time grid cells are identified, which significantly correlate with the mean
monthly SPI time series. These grid cells are subsequently aggregated to predictor regions with similar variability by
means of a Hill-Climbing based k-means cluster analysis. For every large-scale variable and time lag the areal mean
anomalies of those cluster regions are considered as potential predictor variables for a random forest based precipitation
forecast model. All data sets (predictants and predictor variables) are automatically processed for the period from 1948 to
2004. In order to find robust predictor variables for monthly precipitation amounts and to exclude incidental correlations,
the data set is randomly partitioned into two subsets with 27 years of observations each. One of those subsets is utilized
for the cell-wise correlation analysis, the second one is employed for the subsequent calibration of a random forest based
forecast model. Since precipitation usually shows a rather random temporal variability at a monthly time scale, results of
the monthly precipitation forecast are in general unreliable. Thus, modelling results are aggregated to running three-
months precipitation totals. The period from 2005 to 2014 is eventually used for model evaluation.
**2.2 Predictor selection**
As briefly reviewed in the introductory section, seasonal precipitation anomalies in many regions of the world can be
statistically forecasted by means of large-scale atmospheric and oceanic indices or under consideration of customized
parameters. With the aim of automatically deriving adequate predictor variables for monthly precipitation anomalies from
large-scale atmospheric and oceanic conditions an expansive correlation and data-mining procedure is conducted by the
presented seasonal forecast model. A brief summary of global gridded variables which are used for the identification of
potential predictor variables is given in Tab. 1.
In order to reveal the influence of nearby or remote SST anomalies on precipitation characteristics, we make use of the
NOAA Extended global Sea Surface Temperature ERSST V3b (Smith et al., 2008; Smith and Reynolds, 2003), which is
available at a resolution of 2°x2° for the period from 1854 onwards. The data set is based on in situ sea surface temperature
observations only, which are regionalized by means of statistical methods, considering both, low and high frequency
oceanic modes. With the aim of avoiding statistical artefacts, resulting from the variability of the sea ice extent in polar
oceans, we restricted the analysis of SST patterns to the geographical region between 65° N and 65° S. Further we utilize
variables representing the state of the large-scale atmospheric circulation from the NCAR-NCEP reanalysis (Kalnay et
al., 1996). The reanalysis, which is published by the National Center for Environmental Prediction (NCEP) and the
National Center for Atmospheric Research (NCAR), is a near-real-time gridded data set which combines atmospheric





observations with climate modelling results by means of a 4d-assimilation system for the period from 1948 onwards. As
potential atmospheric predictors we used monthly aggregated values of sea level pressure (SLP), the 500 hPa geopotential
height (GPH500) as well as the geopotential thickness between the 500 hPa and 200 hPa pressure level (GPH500-200).
In order to investigate the land surface characteristics and their subsequent effects, we additionally utilize monthly
aggregated global grids of near surface temperature (TEMP), antecedent precipitation amounts (PREC) and snow water
equivalent (SWE) from the NCAR-NCEP reanalysis as potential predictor variables. While the intrinsic pressure related
variables are provided at a spatial resolution of 2.5°x2.5° lat./long., the diagnostic land surface variables are available at
a resolution of approximately 1.875°x1.904°. All predictor fields are cell-wise normalized for every month respectively.
Since the utilized large-scale predictor variables are updated regularly and freely downloadable they are suitable for the
development of an operational seasonal precipitation forecast system.
[Tab. 1]
We assume that typical atmospheric and oceanic indices are determined by large-scale pressure patterns or SST modes
and thus are inherently included in those global gridded data sets. Likewise, additional predictor variables, which might
be specific for a particular target region (e.g. SSTs at adjacent coasts, regional snow cover rates or enhanced water
availability due to high precipitation amounts during previous months) are expected to be covered by the predictor fields
and will be identified as relevant predictors by means of the following correlation and data-mining procedure.
Primarily, based on the first random sample of 27 years, a pearson correlation analysis of the monthly SPI values is
conducted for each gridded large-scale variable and each grid cell. The correlation analysis is separately executed for
every month of the year and for lead times of 1 to 6 months. Thus the identification of relevant predictor variables and
regions is specific for every month and lead time. Particularly for temperature related predictor variables, the time series
might include statistically significant trends, due to anthropogenic greenhouse gas emissions, which frequently exceed
the magnitude of natural variability. However, there is evidence, that seasonal precipitation anomalies in specific target
regions are in fact highly influenced by SST anomalies of nearby or remote oceans, but do not show a distinct response
to global warming during recent decades (Hoerling et al., 2010). Thus the time series of potential predictor grid cells are
detrended prior the correlation analysis. For every variable, each grid cell, which correlates significantly (alpha=0.1) with
the SPI time series is subsequently labeled as potentially predictive for the monthly precipitation forecast. This
comparably low level of statistical significance is deliberately chosen in order to detect second order correlations and
conditional statistical relationships. Overall, the correlation analysis generates a data set of 504 correlation grids, each of
them for a specific predictor variable, month of the year and time lag. Fig. 2 (A1 and A2) exemplarily shows the results
of the correlation analysis for the standardized precipitation values of the Naryn region with global gridded SSTs for
March and September with a lag time of two months.

[Fig. 2]

During March (representing the wet season in Naryn) monthly precipitation shows a clear positive response to January
SST variations in the El Nino core region – a result which has been frequently reported for Central Asia – but also SST
anomalies in the Arabian Sea are highly (positively) correlated. During September (representing rather dry climate
conditions), the ENSO-precipitation relationship is less pronounced and the spatial distribution of potential predictive
SST-regions is rather scattered, indicating less robust statistical relationships.





In a subsequent step each of the correlation grids (which usually contain a large number of potentially predictive grid
cells) is aggregated to a distinct number of correlation regions, by means of a SAGA-GIS based Hill-Climbing k-means
Cluster Analysis (Friedman and Rubin, 1967). Therefore (for every specific month, time lag and predictor variable) the
complete normalized time series of all potentially predictive grid cells within the 56 years of model calibration is
considered. The iterative and unsupervised classification technique firstly randomly allocates every grid cell to one of $k$
clusters. The error sum of squares is calculated as the sum of Euclidian distances of all associated grid cells from the
cluster centroid and displays the quality of the cluster estimation. Every grid cell is subsequently reallocated to the nearest
cluster and cluster centroids and error terms are recalculated. This procedure is iteratively conducted, until the error sum
of squares converges to its minimum value. Basically the clustering algorithm minimizes the error sum of squares within
the cluster groups and maximizes the error sum of squares among them. This leads to definition of regions with similar
temporal variability during the calibration period and thus identifies important large-scale patterns of the considered
predictor variable with high predictive potential for the seasonal precipitation forecast.
As default, the number of clusters for every correlation grid is set to 12, which has been found to adequately identify
typical large scale oceanic and atmospheric features (see e.g. Fig. 2, B1 and B2). An excessive number of clusters might
result in a disjunction of predictor regions, which reduces the predictive skill. On the contrary an insufficient number of
clusters will lead to an aggregation of large regions which might still be characterized by a large inhomogeneity and thus
are not suitable for the derivation of potential predictor variables.
As shown in Fig. 2, the El Nino core region in January (yellow cluster in B1) is identified as one important region for the
forecast of monthly precipitation amounts in March for the Naryn basin, e.g. the positive precipitation anomalies of the
years 1958, 1966, 1969, 1987 and 2004 are all associated with El Nino events. Likewise dry periods usually coincide with
La Nina events, characterized by negative SST anomalies. The nearby clusters, which are characterized by negative
correlations represent compensating currents and are associated with the El Nino Southern Oscillation. The January SST
of the Arabian Sea is identified as an independent predictor variable for the precipitation amounts in March. For the
precipitation variability in September the majority of predictive SST clusters is located in the Pacific Ocean.
The areal mean time series for every cluster are eventually used as potential predictors in the seasonal forecast model.
For all 7 gridded variables the cluster analysis with k=12 clusters is conducted resulting in an overall a number 84 potential
predictors for every month and lead time.
**2.3 Forecast Model Calibration**
For every month of the year and every lead time, one separate statistical forecast model is established based on the
potential predictor variables derived from the correlation and cluster analysis. In order to avoid overfitting and to develop
robust regression relationship, the model calibration is based on the second random sample and thus is independent from
the predictor selection procedure. Some of the potential predictor variables are highly correlated due to their association
to the same phenomenon, e.g. the El Nino Southern Oscillation is manifested in various SST regions and significantly
influences the large-scale pressure and precipitation patterns in many regions of the world. Additionally the distribution
of potential predictor variables is unknown, e.g. precipitation or snow water equivalent are most likely extremely skewed
and not normally distributed. Thus a reliable forecasting approach requires a non-parametric statistical technique, without
any assumption concerning the distribution and statistical independence of predictor variables. We make use of a random
forest based approach (Breiman, 2001), a widely utilized data mining technique, which stands out due to its flexibility
concerning the characteristics of predictant and predictor variables and due to its ability to detect non-linear and
conditional statistical relationships. Basically random forest models represent an advancement of regression tree




algorithms (Breiman et al., 1984) which automatically classify large data sets by means of adequate predictor variables
in order to identify statistical structures in the predictor space, which are highly associated with a response variable
(Gerlitz, 2014; Zorita et al., 1995).
Classification is conducted by means of an iterative procedure. In every processing step one predictor variable and one
split value are identified, which classify the learning sample into two sub groups, characterized by a maximal homogeneity
(i.e. a minimum variance) of the predictant variable. However, since the recursive regression tree approach tends to
considerably overfit the predictor-predictant relationships and does not only classify important structures within the
feature space but also the inherent noise of the predictant variable, the predictive skill of single regression trees is
frequently insufficient. Therefore, random forest applications consider an ensemble of various trees, which are based on
a subset of the complete data set respectively. By means of this bagging approach a large number of trees is constructed.
Prediction values are eventually calculated as the mean of predictions from all single trees. The bagging approach and the
ensemble composition of the final random forest model avoid overfitting and additionally provide an internal error and
confidence estimation (Chen et al., 2012).
The specific forecast models for every month and lead time are constructed based on random forests with 500 realizations.
Regression trees are recursively constructed until the final leaves include 3 observations or less. For the determination of
each splitting criterion, a randomly selected bagging-sample 2/3 of the entire learning sample is utilized.
As the predictant variable the absolute amount of monthly precipitation is used. This allows the subsequent additive
aggregation of the monthly forecast values to seasonal precipitation amounts and the evaluation of the model at different
temporal scales. Fig. 3 shows exemplarily the results of the monthly precipitation forecast with varying lead times for the
Naryn catchment for the evaluation period from 2005 to 2014.

[Fig. 3]

Values are converted to the monthly standardized precipitation index based on observations from the entire model
calibration period from 1948 to 2003. Obviously the variability of precipitation amounts is highly underestimated by the
random forest based precipitation forecast models, which is a typical feature of regression based statistical models,
particularly if the predictant variable is  characterized by a large non-predictable noise. Furthermore, the correlation of
forecasted and observed precipitation is low with values below 0.2 for most months and lead times. The rather poor results
at the monthly scale certainly reflect the non-predictable noise of monthly precipitation amounts and thus lead to the
assumption that modelling results should not be evaluated based on discrete monthly values due to the high frequency
variability of precipitation events. This is confirmed by the aggregation of observations and modelling results to three-
months running totals, which leads to a significant increase of correlation and a decrease of variance underestimation.
Fig. 4 shows the time series of SPI values for running three-months total precipitation amounts. Particularly the wet winter
and spring seasons of 2004/2005 and 2009/2010 in Naryn are satisfactory predicted by the statistical forecast models with
varying lead times, but also the negative precipitation anomalies of the 2008 and 2010 spring seasons are well captured.
The best predictive skill is found for a lead time of 1 to 3 months ($r^2 > 0.3$), however, even for a lead time of 6 month a
certain skill is detected ($r^2 = 0.13$).

[Fig. 4]





With this in mind we define two forecast periods with a length of three-months respectively. For every particular month,
one forecast model is established based on the sum of monthly modelling results of the upcoming three-months, a second
forecast is conducted based on lead times of 4 to 6 months:

$$F[1:3] = \sum_{l=1}^{3} RF(m,l) \quad \& \quad F[4:6] = \sum_{l=4}^{6} RF(m,l)$$

where $RF(m,l)$ is the specific Random Forest forecast model based on predictor variables of the month $m$ and precipitation
anomalies occurring after a lead time of $l$ months.
The evaluation of the seasonal forecast model is automatically conducted for the running three monthly precipitation
totals.
**2.4 Model Evaluation**
Since the skill of the automatic forecast model is likely to vary depending on the target area and the associated
precipitation regimes during different seasons, an evaluation of the automatic seasonal forecast model performance is
necessary in order to assess the reliability of the forecast and to interpret the results. Based on the independent period
from 2005 to 2014, the deterministic forecasts of three-months running totals are automatically evaluated. Although, the
analysis based on 10 years only might be insufficient for the precise quantification of statistical model skill, we assume
that a general assessment of the model quality is feasible. We abstained from the implementation of a cross-validation
procedure due to the high computational demands of the predictor selection routine. For each of the running three-months
periods, traditional performance parameters such as correlation, bias and root mean square error (RMSE) are computed,
which enables the assessment of the model performance for various seasons. In order to achieve a maximal comparability
of different target areas, bias and RMSE are specified as the percentage of the long time precipitation totals for each three
month period respectively.
Moreover, since stakeholders often require robust predictions of anomalous periods, the ability of the forecast model to
forecast drought and moist conditions is evaluated by means of receiver operating characteristics (ROC) for each three-
months period and areas under the curve (AUC) are provided. Therefore the running three-months precipitation totals are
converted to the associated standardized precipitation indices, based on observations of the entire model calibration period.
The deterministic SPI forecast is then converted into a probabilistic prediction by means of a simple residual based
approach. Assuming that SPI residuals are normal distributed for each three-months period respectively, we use the 10
years of independent observations to assess the specific standard deviation for each of the three-months periods, which is
subsequently utilized to transform the deterministic forecast into a normalized probability distribution. ROC curves are
then constructed for SPI threshold values of -0.5, representing moderate drought, and +0.5, indication moist conditions.
For various probability thresholds positive hit rates (defined as the number of correctly identified droughts divided by the
overall number of drought events) are plotted against the false negative rate (defined as the coefficient of the number of
false alarms and the number on non-drought conditions). ROC curves for moist conditions are equivalently constructed.
Eventually the area under the curve is interpreted as a performance measure of the seasonal forecast model. AUC-values
near 1 indicate a perfect predictive skill considering the forecast of droughts or moist periods, values of 0.5 or less indicate
no predictive skill at all.




**3    Model Application to headwater catchments in Central and South Asia**
With regard to an increasing demand of climatological and hydrological forecasts in this vulnerable region, we applied
the presented model to four headwater catchments covering different climatic settings in Central and South Asia (see Fig.
4). The Naryn catchment is located in the Central Tian Shan region, drains towards west and is one of the major tributaries
of the Syr Darya river. Kokcha is situated in the Hindu Kush region in North Eastern Afghanistan and contributes to the
Amu Darya river system. As presented by the mean 850 hPa wind field of the NCAR reanalysis (Fig. 5), both catchments
are mainly controlled by extratropical westerly circulation patterns (with contributions from south during winter and from
high latitudes during summer) and receive a precipitation maximum during spring season. Particularly Kokcha is
characterized by a large inter-annual precipitation variability as reflected by the coefficient of variation (defined as the
standard deviation of precipitation amounts divided by the long-term mean for every month) above c=1 during dry season.
Due to the high elevation, precipitation during moist season mainly falls as snow and is released during warm and dry
summer season (Barlow and Tippett, 2008; Dixon and Wilby, 2015; Schär et al., 2004). Thus winter and spring
precipitation amounts in the mountainous areas provide a vast share of the main Central Asian river flow during the
vegetation period and form the basis for the irrigation dependent agriculture of the riaparian countries, which are
characterized by semi-arid to arid climate conditions throughout the year.
In contrast the catchments of Karnali and Arun are both located in the Nepal Himalayas and represent two important
tributaries of the Ganges river. During winter, both catchments are under influence of westerly winds and receive a certain
amount of precipitation due to the passage of westerly disturbances, however the maximum of precipitation is associated
with the Indian Summer Monsoon, which transports moist air masses from the Arabian Sea and the Bay of Bengal into
the target areas. A correlation analysis of seasonal SPI values for the selected target areas (not shown) reveals their
association to the described pluviometric regimes. In general significant correlations (alpha<0.1) are found for Naryn
with Kokcha and Karnali with Arun throughout the year, which indicates their association to the continental Central Asian
and the Monsoonal influenced subdomain respectively.
[Fig. 5]
The model application to the selected headwater catchments with different climatic characteristics enables the
identification of important predictor variables and the analysis of the model performance for the varying pluviometric
regimes of the Central and South Asian domain. In the following section we briefly introduce the large-scale atmospheric
processes which lead to a spatial and seasonal differentiation of precipitation amounts in this vast target domain and
present some influencing factors which have been frequently linked to the inter-annual precipitation variability.
Subsequently we discuss the modelling results with regard to major large-scale atmospheric forcing mechanisms and
provide a sensitivity analysis which uncovers important influencing factors on the precipitation variability under
consideration of different seasons of the year.

**3.1 Pluviometric regimes and precipitation variability over Central and South Asia**

In general the climate of Central and South Asia is influenced by two major pluviometric regimes, which are related to
westerly and monsoonal circulation systems. During boreal cold season the entire region is influenced by westerly



circulation patterns and precipitation is mainly associated with mid-latitude disturbances originating over the Atlantic
Ocean and the Mediterranean (Bohner, 2006; Bothe et al., 2011; Gerlitz et al., 2015; Maussion et al., 2014). Since the
track of westerly disturbances is mainly determined by the position of the 200 hPa westerly Jetstream at the polar frontal
zone, a clear seasonal cycle of precipitation is distinctly defined. Particularly the western part of the Himalayas receive a
considerable amount of winter precipitation associated with the uplift of westerly air masses, which reaches up to 60 %
of the annual precipitation total (Bohner, 2006; Gerlitz et al., 2015; Wulf et al., 2010). During spring the zone of westerly
precipitation migrates towards north, reaches the Hindu Kush region and the Pamir in March and continues to the Tien
Shan region in May. Mariotti (2007) show that during winter season, a northward current over the Arabian countries
transports tropical air masses into Central Asia, which represents an important moisture source for the westerly air masses.
While the continental Central Asian countries remain under influence of extratropical westerly air masses throughout the
year, the tropical monsoon circulation is established over South Asia during summer season (Bohner, 2006; Bookhagen
and Burbank, 2006; Gerlitz et al., 2015). Due to a declining strength of the monsoonal moisture fluxes towards west, clear
gradient of precipitation totals from east to west has been observed (Bohner, 2006; Wulf et al., 2010).
Investigations of the inter-annual variability of precipitation rates over Central and South Asia have frequently been
conducted. Most studies (Li and Yanai, 1996; Peings and Douville, 2009; Prodhomme et al., 2014) show evidence that
the intensity of the Indian Summer monsoon is associated with the magnitude of pressure gradients between the Indian
Ocean and the Asian continent, which has been linked to the extent of the snow cover over the Asian mainland and the
SST of the Indian Ocean (Wu and Qian, 2003). Moreover, many studies highlight the importance of the Southern
Oscillation for the intensity of monsoonal precipitation. Studies by Pokhrel et al. (2012) and Sigdel and Ikeda (2013)
indicate that El Niño events lead to reduced moisture fluxes into South Asia. Ashok et al. (2001) further identified the
Indian Ocean Dipole as an important predictor for the Indian Summer Monsoon. Some studies illustrate that the
correlation of the Southern Oscillation index (SOI) and the Indian Summer Monsoon precipitation is non-stationary and
weakened during recent decades (Kumar et al., 1999; Wang and He, 2012). However, Yim et al. (2013) detected a recovery
of the negative ENSO-Monsoon relationship during the 1990s. Chang et al. (2001) suggested that the breakdown of robust
relationships is due to changes in the North Atlantic climate. Rajeevan et al. (2006) detected a statistically significant
correlation of Western Europe winter temperatures and subsequent monsoonal precipitation amounts.
In contrast for the variability of winter and spring precipitation (associated with westerly weather patterns over Central
and South Asia) a positive relationship with the El Nino Southern Oscillation has been observed. Severe droughts have
been linked to the El Nino cold phase (La Nina) (Barlow et al., 2002, 2015; Hoell et al., 2013). Roghani et al., 2015 and
Shirvani and Landman, 2015 found statistically significant correlations of the Southern Oscillation index during summer
and autumn with precipitation amounts over Iran in subsequent winter. Likewise, a significant positive correlation of the
ENSO state with winter precipitation amounts over the Southern Himalayan slopes has been detected (Dimri, 2013; Yadav
et al., 2010). Mariotti (2007) show that the moisture fluxes originating over the Arabian Sea are enhanced during ENSO
warm phase due to the strengthening of the southwesterly current over the Arabian countries. Beside tropical SST modes,
the impact of Northern Atlantic climatic conditions on the winter climate of Central Asia have been frequently
investigated. Bothe et al. (2011) show that drought and moist winter seasons over Central Asia are dominated by different
wave patterns over the Eurasian sector. Schiemann et al. (2008) report that an anomalous location or a decreasing strength
of the westerly Jetstream result in drought conditions over parts of Central Asia due to modified tracks and intensities of
westerly disturbances. Dimri (2013) found that a distinct southward shift of the westerly Jetstream is associated with wet
winter conditions over the Himalayas. Syed et al. (2006, 2010) indicate that positive winter precipitation anomalies over
Afghanistan, Pakistan and Tajikistan are usually associated with El Nino events combined with a positive state of the
North Atlantic Oscillation. Negative correlations between the NAO index and observed precipitation anomalies were





found for Kyrgyzstan and Northern Uzbekistan. Yin et al., (2014) further showed that the positive phase of the Eastern-
Atlantic/Western-Russia and the Polar/Eurasian patterns lead to enhanced moisture fluxes into Central Asia.
Eventually, Hartmann et al. (2016) suggest that beside of well-known atmospheric modes, the sea surface temperatures
of the main moisture sources might influence the precipitation climate of the Tarim basin.
**3.2 Modelling results**
The seasonal precipitation model, including the automatic predictor selection routine, has been applied to each of the
selected target regions and the results have been evaluated with regard to different seasons and the accompanying
precipitation regimes. Fig. 6 shows the time series of observed three-months running totals (blue bars) and the aggregated
results (red lines) of the F[1:3] and the F[4:6] forecast model for the evaluation period from 2005 to 2014. In order to
keep the annual cycle, values are displayed at the center of each three month period. The date of forecast generation is
1.5 months earlier for F[1:3] and 4.5 months earlier for F[4:6]. The corresponding SPI values for each of the running
three-months periods are presented and the 90% confidence interval of the residual based probabilistic forecast is
illustrated. Tab. 3 summarizes the modelling results in terms of correlation, bias, RMSE and AUC for moderate drought
and moist conditions. The performance measures are provided for each of the running three-months periods, respectively.
Particularly for the Naryn catchment in Kyrgyzstan, drought and moist conditions during the evaluation period are well
captured by the statistical model. Moist spring seasons in 2005, 2010 and 2013 are adequately predicted by both, the
F[1:3] and the F[4:6] forecast models. Also the drought years of 2008 and 2012 are accurately predicted by the forecast
model, although the severity of the extreme 2008 drought is highly underestimated. Correlations between observed and
modelled precipitation totals are high (>0.4) for the prediction of winter and spring precipitation. AUC values >0.7
indicate that the model is capable to forecast moderate drought and moist conditions during winter and spring.  For the
dry summer season the model fails to adequately capture the inter-annual variability of precipitation amounts, which is
evidenced by negative correlations and RMSE values up to 44 % of the long term mean.
The headwater catchment of Kokcha in Northen Afghanistan shows a similar variability of precipitation forecast with
positive SPI values during 2010 and negative precipitation anomalies during 2008 and 2011, which indicates a common
large-scale climatic forcing of these Central Asian catchments. The drought conditions during boreal cold seasons of
2007/2008 and 2010/2011 are evident in both the observational and the modelled time series. However the variability of
precipitation rates in Kokcha is highly underestimated by the statistical model, which leads to higher RMSE values
throughout the year. Accordingly the AUC values seldom exceed 0.7 and indicate a limited skill of the model to forecast
drought or moist seasons for Kokcha during the evaluation period.
For the monsoonal influenced catchment Karnali in Western Nepal maximum correlations were achieved during winter
and summer. Particularly for the monsoon season, high correlations and AUC values above 0.7 result in both the F[1:3]
and the F[4:6] forecasts, which indicates the ability of the model to predict monsoonal drought periods several months in
advance. The negative precipitation anomalies during summer monsoon seasons of 2007 and 2009 are captured by the
forecast model. However the severe monsoonal drought in 2009 is strongly underestimated. Likewise the extreme winter
drought of 2008/2009 in Karnali and Arun is only rudimentarily captured by the F[1:3] forecast model. For the transition
seasons, negative correlations, high RMSE values of up to 60% of the long-term mean and AUC values below 0.5 indicate
a poor performance of the statistical model. The performance of the F[4:6] model is limited for the Karnali catchment and
shows moderate correlations and AUC values only for the summer monsoon season. Eventually for the catchment of Arun
in Eastern Neapal notable correlations of observed precipitation amounts were only detected for F[1:3] based prediction
of the monsoon season. For winter and spring season the skill of the model is fairly limited.





In overall, the statistical model adequately captures the variability of westerly precipitation amounts for the Naryn,
Kokcha and Karnali catchments. For the monsoonal influenced target areas Karnali and Arun a certain predictability of
summer precipitation amounts was detected. During dry seasons (summer and autumn in Central Asia and transition
seasons in Nepal), the model performance for the evaluation period is generally poor.
[Fig. 6]
[Tab. 2]
**3.3 Sensitivity Analysis**
In comparison to e.g linear models with a small set of independent predictor variables, the complex structure of the
presented random forest based statistical model does not directly reveal physically interpretable input–output relationships.
Particularly the fact that the predictor selection procedure generates a large sample of partially highly correlated predictor
variables, which basically comprise the same information concerning the large scale climatic variability, impedes a direct
interpretation of the predictor importance and variable response. The aggregation of the modelling results to three-months
running precipitation totals further complicates the analysis of the inherent model structure. In order to overcome the
blackbox character of the statistical model, we conducted a sensitivity analysis for the selected target areas under
consideration of well-known atmospheric indices. In order to discover the influence of selected climate indices, individual
random forest models were forced with modified input data, containing only those predictor variables, which are highly
correlated with the considered index. This enables the estimation of the fractional response of the model to the considered
parameter and reveals their influence on the hydroclimatic variability in the selected target areas. The results of the
sensitivity analysis further enable the comparison of the modelling results with previous studies which utilize traditional
climate indices in order to explain or forecast precipitation anomalies in Central and South Asia (see section 3.1 for a
brief summary).
In order to estimate the influence of selected climate indices for different seasons, we analyzed the F[1:3] forecast models
based on predictor variables from November, February, May and August, which enable the precipitation forecast for
December to February, March to May, June to August and September to November respectively. Each F[1:3] forecast
model consists of 3 random forests, with lead times between one and three month, which are all based on predictor
variables derived from globally gridded variables from one particular month. With the aim of investigating the model
response to a selected climate index, the time series of potential predictor variables, which are significantly correlated
with the indice (alpha=0.01) are maintained, while the others were set to zero. The statistical forecast model is then applied
with modified predictor time series. The results are converted to three-months SPI values and are plotted for the entire
calibration and evaluation period (see Fig 7) in comparison with the evolution of the parameter itself. Particularly the
response of the model to selected indices during drought and moist periods (represented by red to blue bars in Fig.7)
provide an insight into the internal model structure and enables the identification of important large-scale influencing
factors triggering anomalous hydroclimatic conditions during the subsequent season. However, due to the non-linear
nature of the statistical model, the response fractions should not be perceived as independent or additive and should rather
be interpreted as a general sensitivity of the model.
As potentially important large scale climate indices we make use of the index of the North Atlantic Oscillation (NOA)
and the El Nino-3 Index, which are frequently mentioned as important influencing factors on the Central and South Asian



precipitation climate (Barlow et al., 2002; Hoell et al., 2013; Khidher and Pilesjö, 2014; Syed et al., 2006). The results of
the sensitivity analysis for NAO and ENSO are plotted in Fig 7.
Further we tested the sensitivity of the model to variations of the Eastern-Atlantic/Western-Russia (EA/WR) and the
Polar/Eurasian (P/E) pattern, as proposed by Yin et al., (2014). Additionally we utilized SST anomalies at adjacent oceans
as suggested by Hartmann (2015), which were extracted from the ERSST V3b data set. Since the response to most indices
was found to be negligible, we refrain from plotting the results of the sensitivity analysis, but will nevertheless highlight
some findings.
The plotted time series (Fig. 7) clearly indicate that the state of the El Nino Southern Oscillation determines the
precipitation variability in the entire target area. For winter and spring season (represented by the F[1:3] forecast models
for November and February) a positive response of the model to predictors related to the ENSO-3 index is evident for all
target areas, indicating an intensification of moisture fluxes and associated westerly disturbances over the entire domain
during ENSO warm phase and a reversed effect during cold phase of El Nino, which is consistent with previous studies
on the variability of cold season precipitation totals in the vast target domain (Barlow et al., 2002, 2015; Dimri, 2013;
Mariotti, 2007; Syed et al., 2006). The model response is strongest for the moist seasons, which is Mar-Apr-May for the
Central Asian and Dec-Jan-Feb for the Himalayan catchments. This coincides with a moderate to high model performance
for Naryn and Kokcha during spring and Karnali during winter season and emphasizes the relevance of ENSO for the
winter and spring moisture fluxes into Central and South Asia. It is obvious that particularly during winter and spring
drought conditions frequently occur simultaneously in Naryn and Kokcha and are mostly associated with an ENSO cold
phase. Winter drought conditions in the Himalayan headwater catchments likewise occur more often subsequent to La
Nina events. For example the frequently investigated prolonged winter and spring drought between 1999 and 2001 in the
entire target domain is obviously associated with a cold phase of the El Nino Southern Oscillation. Likewise the recent
negative precipitation anomalies during the 2007/2008 winter for the entire target domain and the severe 2008/2009 winter
drought over Nepal, are both related to a cold phase of ENSO, which is well captured by the winter and spring forecast
models.
Moist winter and spring conditions (e.g. during 2002, 2003 and 2005 in the entire target) occur frequently under ENSO
warm phase.
[Fig. 7]

In addition, the winter and spring precipitation forecast models for Naryn and to a lesser extent for Kokcha distinctly
responds to variations of predictor variables related to the North Atlantic Oscillation. Particularly during spring, which
represents the moist season in both headwater catchments the model obviously shows a negative reaction to variations of
the NAO index, which is of the same magnitude as the response to ENSO related predictors. Again this is confirmed by
previous studies of Syed et al. (2006) who found a negative correlation between the state of the NAO and winter precip-
itation over the Northern Central Asian countries and a reverse relationship for a band covering Iran, southern Afghanistan
and Pakistan. Thus the combination of a negative NAO phase with a warm phase of the El Nino Southern Oscillation is
likely to trigger drought conditions over Central Asia in spring. During winter and particularly for the target areas in the
Himalayan region, the response to NAO related predictor variables is distinctly less defined. For the Naryn basin a slight
positive response of winter precipitation amounts to variations of the NAO index during autumn exists. Further a positive
response to the EA-WR is evident for winter precipitation amounts in the Naryn catchment, however the magnitude of
the model response is substantially lower in comparison with ENSO and NAO (not shown).





Variations of adjacent Sea Surface Temperatures (not shown) tend to play a less significant role for the precipitation
variability in all target areas during winter and spring season. Solely the Mar-Apr-May forecast model for Naryn shows
a distinct positive response to SSTs of the Bay of Bengal. However, the simultaneous occurrence with ENSO related
anomalies suggests connection of both parameters, most likely due to the statistical dependence of the El Nino Southern
Oscillation and the Indian Ocean Dipole, which dominates the SST variations in the Indian Ocean. During summer and
autumn season, the positive response of the F[1:3] model to El Nino events remains constant for the target areas Naryn
and Kokcha, which are under influence of westerly circulation patterns throughout the year. Although model performance
for the Central Asian headwater catchments is poor during summer season, the results of the sensitivity study are con-
sistent with findings by Mariotti (2007), who proposed seasonal independent enhanced south easterly moisture fluxes into
Central Asia during the ENSO warm phase.
For the monsoonal influenced catchments Karnali and Arun, a distinct negative relationship between ENSO variations
and summer and autumn precipitation variability is evident in the modelling results, which again is confirmed by previous
studies (Rajeevan and Pai, 2007; Sigdel and Ikeda, 2013; Wu et al., 2009). Also the analysis suggested response of the
summer monsoon to SST variations in the Indian Ocean (not shown) which, albeit, is of significant smaller magnitude.
No response of the monsoonal precipitation forecast was found to the extratropical NAO, EA/WR and P/E signals.
**4    Summary and Outlook**
We presented a statistically based modelling framework, which automatically identifies suitable predictors from globally
gridded climate variables by means of an extensive data mining procedure and explicitly avoids the utilization of typical
large-scale climate indices. This leads to an enhanced flexibility of the model and enables its automatic calibration for
any target area without any prior assumption concerning adequate predictor variables. Potential predictor variables are
derived by means of a cellwise correlation analysis of precipitation anomalies within a user selectable target area with
global climate variables. The correlation analysis is conducted for monthly values with lead times ranging from one to
six months. For each potential predictor variable, month and lead time, significantly correlated grid cells are aggregated
to predictor regions by means of a variability based cluster analysis. Finally for every month and lead time, an individual
random forest based forecast model is constructed, by means of the preliminary generated predictor variables. In order to
reduce the risk of overfitting, predictor selection and model calibration are based on independent samples. Due to the
large noise of observed precipitation amounts at a monthly time scale, the random forest based forecasts for every month
with lead times of one to three-months and four to six months are aggregated to running three-months predictions. These
are automatically evaluated based on an independent period and modelling performance measures are provided for each
of the running three-months predictions, which enables the assessment of the model performance for different seasons of
the year.
The model was exemplarily applied to selected headwater catchments in Central and South Asia. While the Central Asian
catchments are under influence of westerly air masses throughout the year, the target areas in Southern Asia receive
moisture fluxes from westerly winds during winter and are under influence of the South Asian Monsoon during summer
season
Particularly for the catchment of Naryn in the Tien Shan region (Kyrgyzstan) but also for Kokcha in the Hindu Kush
(Northern Afghanistan) the model showed a solid performance, especially for the moist spring season. The capability of
the model to predict moderate drought events or anomalous moist conditions up to six months in advance is reflected by
high correlations with observations and AUC values >0.7 for the evaluation period. Due to the fact that precipitation in



the high elevations mainly falls as snow and is released during dry summer season, the irrigated agriculture of the downstream countries is highly vulnerable to drought events during winter and spring. Some studies indicate, that the natural summer discharge of the tributaries of the major Central Asian rivers can be accurately forecasted by means of winter precipitation amounts or snow cover rates, which are usually available in spring (Barlow and Tippett, 2008; Dixon and Wilby, 2015). A modelling chain including statistical precipitation forecasting und runoff prediction could extent the forecast range and enable the initiation of adequate adaption strategies.

For the South Asian target areas the model performance was found to be slightly lower, but particularly for the economically important monsoonal precipitation amounts an adequate skill could be detected.

A sensitivity analysis of the complex statistical model by means of well-known climate indices shows, that the model automatically finds relevant predictor variables, among others, those which are associated with typical climatic modes, such as the North Atlantic or the El Nino Southern Oscillation. Further the sensitivity analysis enables the estimation of the model response to specified climatic modes and thus reveals the major influencing factors for the observed precipitation variability. The winter and spring precipitation amounts in the entire target were found to be highly determined by the state of the El Nino Southern Oscillation with positive precipitation anomalies during El Nino events in the entire target domain. Additionally for the Central Asian catchments the state of the North Atlantic Oscillation has been identified as an important controlling factor. The sensitivity analysis of the model suggests, that drought events are frequently triggered by a positive NAO phase in combination with an ENSO cold phase. Concerning the forecast of summer precipitation amounts for the monsoonal influenced regions Karnali and Arun, the model shows a distinct negative response to El Nino events.

In general, the statistical model is characterized by a large underestimation of variance, but the forecast of a certain drought risk appears feasible. The accurate prediction of severe drought periods however remains difficult by means of statistical techniques. Therefore the atmospheric and oceanic patterns which trigger extreme drought or moist conditions and the interaction of potential influencing factors (such as the state of the North Atlantic Oscillation, the El Nino Southern Oscillation or SST anomalies at adjacent coasts) need to be further investigated.

**Acknowledgements**

This work was carried out within the framework of the CAWa (Water in Central Asia) project (http://www.cawa-project.net, contract no. AA7090002), funded by the German Federal Foreign Office as part of the "Berlin Process".

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



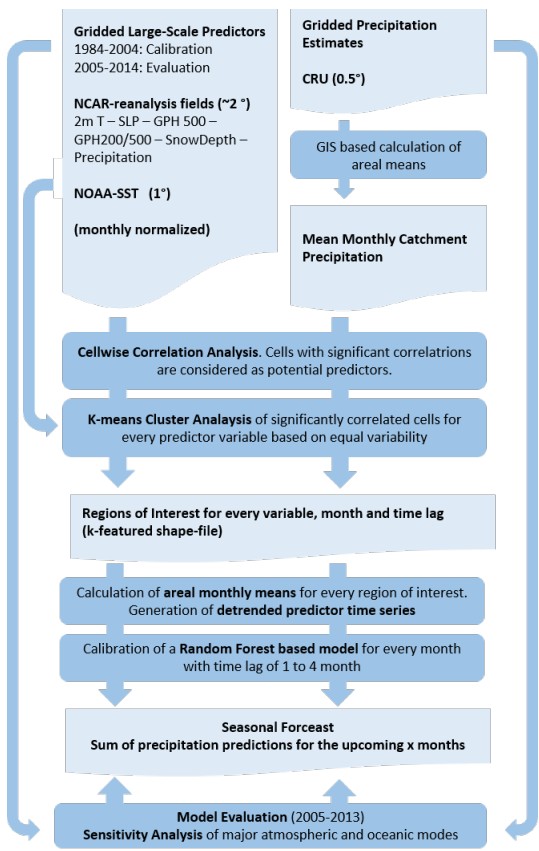

**Fig 1: Flow Chart representing the major components of the seasonal forecast model.**





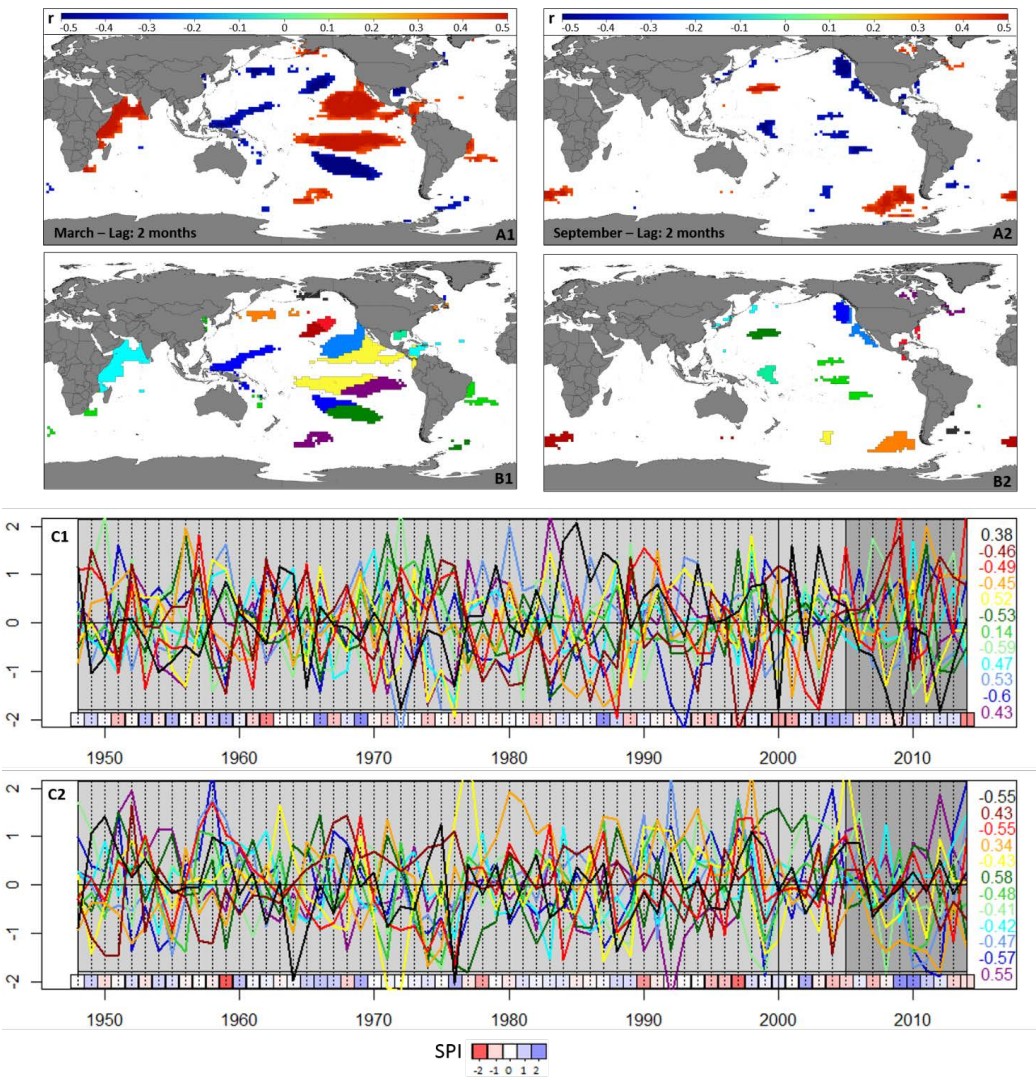

**Fig.2: Exemplary representation of SST grid cells which are significantly correlated with precipitation anomalies of the Naryn basin for March and September with a lead time of two months (A1 and A2) and the aggregation to predictor regions based on the Hill Climbing k-means cluster analysis (B1 and B2). The diagrams (C1 and C2) show the time series of z-normalized mean SSTs during the selected months for each of the cluster regions (same color) and the subsequent hydroclimatic variations in the Naryn catchment (expressed as red to blue rectangles, indicting SPI values between -2 to 2). The colored values on the right indicate the correlation of mean cluster SST anomalies and the corresponding SPI values.**



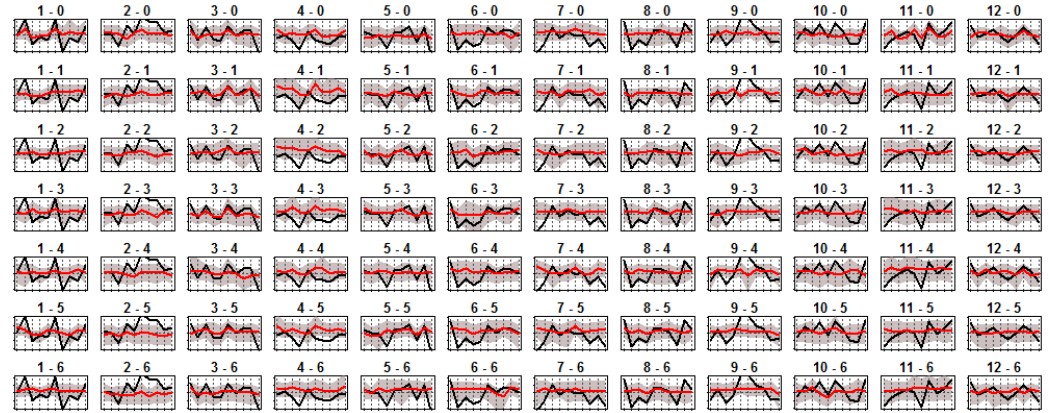

**Fig. 3: Results of the random forest based monthly precipitation forecast models (red) and observations (black) for the Naryn catchment for the period from 2005 to 2014 (x-axis). Values are displayed as monthly SPI values between -2 and 2. The numbers indicate the month for which the forecast is conducted and the particular lead time (e.g. 1-1 shows the results for January precipitation based on predictor variables from December). The shaded area indicates the range of prediction values of all single tree models belonging to the random forest forecast model.**

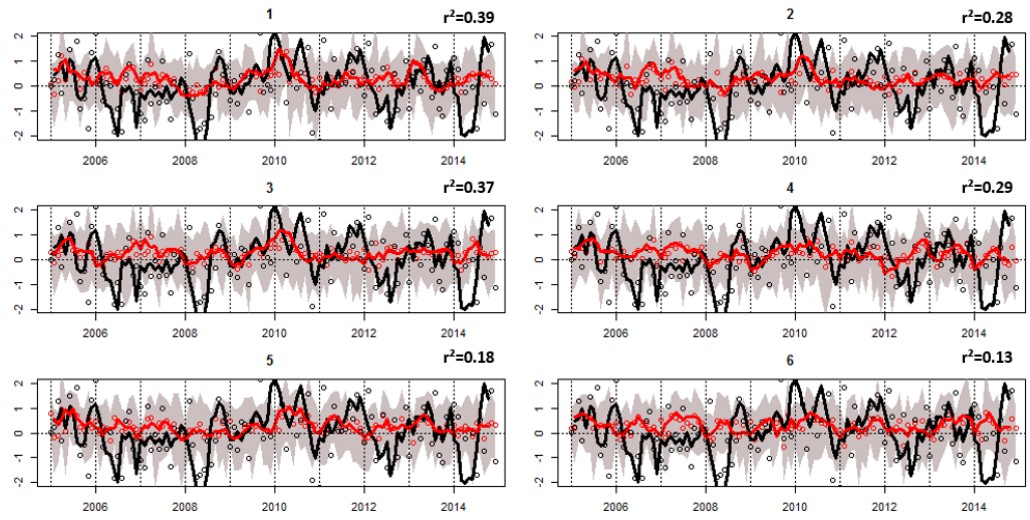

**Fig. 4: Time series of observed (black line) and forecasted (red line) running three-months SPI values for the Naryn catchment. The shaded areas indicates the three month total of maximum and minimum forecasts of single trees of the random forest model.**
**Points show the observations and forecast at a one-monthly scale.**




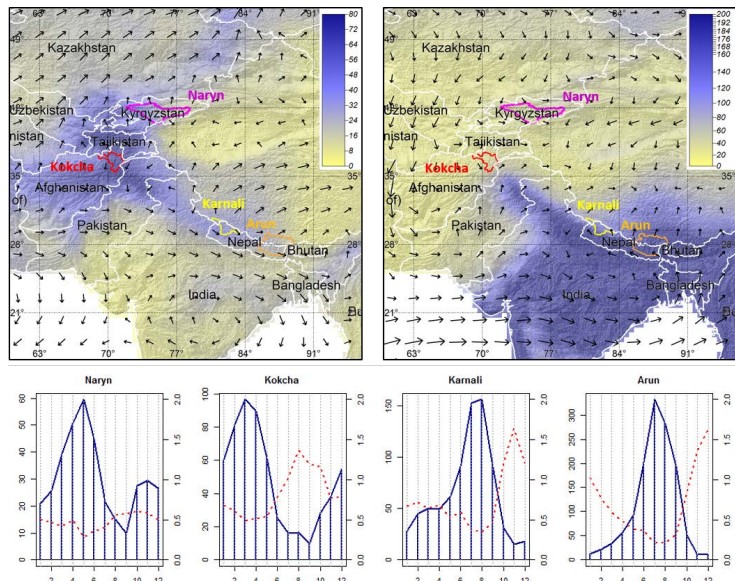

**Fig. 5: Location of selected target areas as well as the mean precipitation total [mm] (CRU-TS) and the mean 850 hpa wind field (NCEP-NCAR) during DJF (left) and JJA (right).**

**Diagrams show the mean monthly precipitation amount for every catchment in mm (blue bars) as well as the Coefficient of Variation (red line)**



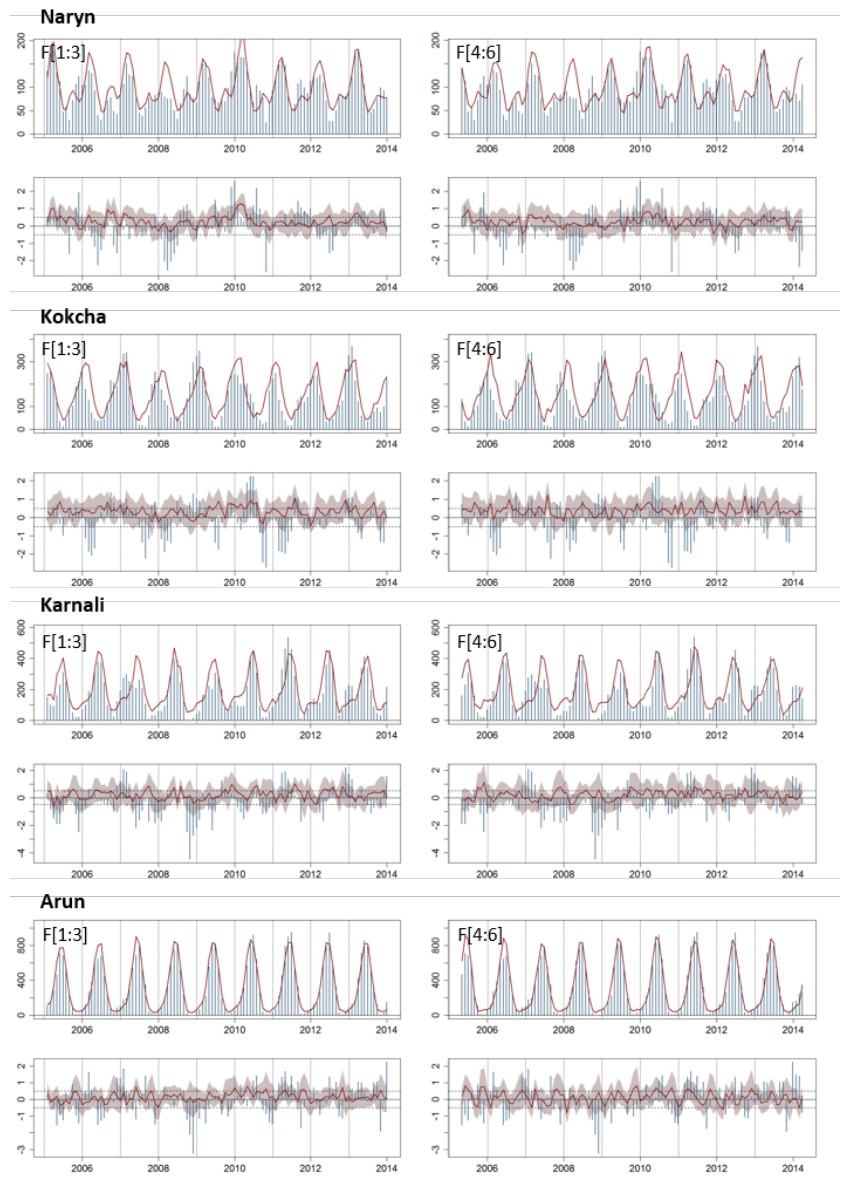

**Fig. 6: Observed running three-months precipitation totals (blue bars) and modelling results (red line) of the F[1:3] and the F[4:6] model for Naryn, Kokcha, Karnali and Arun. The upper panels show absolute precipitation totals for running three-months preiods, the lower panel show the corresponding SPI index for each three-months period respectively. Shaded areas indicate the 90% interval of the residual based probabilistic forecast.**





Fig. 7: Response of the F[1:3] model to predictors associated to ENSO (red) and NAO (green) for different seasons and target areas (upper panels of each diagram). Time series of the considered climate indices (lower panel) for the month of forecast generation.





| Acronyme | Variable Name | Unit | Source | Spatial Resolution |
|---|---|---|---|---|
| SST | Sea Surface Temperature | °C | ERSST V3b | 2°x2° |
| SLP | Sea Level Pressure | hPa | NCAR-reanalysis | 2.5°x2.5° |
| GPH500 | Geopotential Height at 500 hPa | m | NCAR-reanalysis | 2.5°x2.5° |
| GPH500-200 | Geopotential Thickness between 500 and 200 hPa | m | NCAR-reanalysis | 2.5°x2.5° |
| TEMP | Near Surface Temperature | °C | NCAR-reanalysis | 1.875°x1.904° |
| PREC | Previous Precipitation | mm | NCAR-reanalysis | 1.875°x1.904° |
| SWE | Snow Water Equivalent | mm | NCAR-reanalysis | 1.875°x1.904° |

**Tab. 1: Globally gridded variables utilized as potential predictor variables by the statistical forecast model**

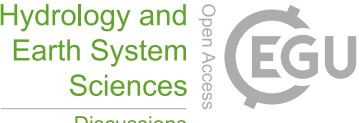

**Naryn**

| Month | Forecast Range | COR | BIAS[%] | RMSE[%] | AUC(-0.5) | AUC(0.5) | Month | Forecast Range | COR | BIAS[%] | RMSE[%] | AUC(-0.5) | AUC(0.5) |
|---|---|---|---|---|---|---|---|---|---|---|---|---|---|
| 1 | 2-4 | 0.71 | 0.07 | 0.13 | 1.00 | 0.57 | 1 | 5-7 | 0.29 | 0.09 | 0.20 | 0.56 | 0.55 |
| 2 | 3-5 | 0.68 | 0.16 | 0.20 | 1.00 | 0.86 | 2 | 6-8 | 0.50 | 0.12 | 0.26 | 0.64 | 0.83 |
| 3 | 4-6 | 0.70 | 0.16 | 0.20 | 0.80 | 0.86 | 3 | 7-9 | 0.50 | 0.10 | 0.28 | 0.94 | 0.78 |
| 4 | 5-7 | 0.58 | 0.07 | 0.18 | 0.83 | 0.45 | 4 | 8-10 | -0.15 | 0.12 | 0.44 | 0.43 | 0.25 |
| 5 | 6-8 | 0.53 | 0.10 | 0.25 | 0.86 | 0.83 | 5 | 9-11 | 0.35 | 0.04 | 0.31 | 0.86 | 0.56 |
| 6 | 7-9 | 0.08 | 0.07 | 0.31 | 0.56 | 0.39 | 6 | 10-12 | -0.26 | 0.02 | 0.24 | -- | 0.36 |
| 7 | 8-10 | -0.31 | 0.13 | 0.44 | 0.21 | 0.30 | 7 | 11-1 | -0.10 | 0.07 | 0.32 | 0.14 | 0.45 |
| 8 | 9-11 | -0.22 | 0.02 | 0.38 | 0.50 | 0.22 | 8 | 12-2 | 0.66 | 0.26 | 0.42 | 0.88 | 0.55 |
| 9 | 10-12 | -0.28 | 0.05 | 0.25 | -- | 0.36 | 9 | 1-3 | 0.53 | 0.20 | 0.40 | 0.38 | 0.50 |
| 10 | 11-1 | -0.19 | 0.03 | 0.35 | 0.36 | 0.65 | 10 | 2-4 | 0.39 | 0.08 | 0.18 | 0.64 | 1.00 |
| 11 | 12-2 | 0.23 | 0.22 | 0.41 | 0.25 | 0.70 | 11 | 3-5 | 0.46 | 0.17 | 0.25 | 0.78 | 0.63 |
| 12 | 1-3 | 0.86 | 0.16 | 0.31 | 0.75 | 0.72 | 12 | 4-6 | 0.50 | 0.18 | 0.24 | 0.90 | 0.88 |

**Kokcha**

| Month | Forecast Range | COR | BIAS[%] | RMSE[%] | AUC(-0.5) | AUC(0.5) | Month | Forecast Range | COR | BIAS[%] | RMSE[%] | AUC(-0.5) | AUC(0.5) |
|---|---|---|---|---|---|---|---|---|---|---|---|---|---|
| 1 | 2-4 | 0.27 | 0.07 | 0.25 | 0.64 | 0.44 | 1 | 5-7 | 0.39 | 0.22 | 0.44 | 0.95 | 0.50 |
| 2 | 3-5 | 0.30 | 0.26 | 0.35 | 0.55 | 0.75 | 2 | 6-8 | -0.19 | 0.12 | 0.67 | 1.00 | 0.13 |
| 3 | 4-6 | 0.45 | 0.21 | 0.33 | 0.80 | 0.43 | 3 | 7-9 | -0.06 | 0.10 | 0.80 | 0.65 | 0.64 |
| 4 | 5-7 | 0.54 | 0.16 | 0.43 | 0.75 | 0.79 | 4 | 8-10 | -0.18 | 0.32 | 0.48 | 0.50 | 0.57 |
| 5 | 6-8 | 0.46 | 0.15 | 0.58 | 0.38 | 1.00 | 5 | 9-11 | 0.53 | 0.19 | 0.37 | 0.79 | 0.64 |
| 6 | 7-9 | 0.41 | 0.10 | 0.73 | 0.60 | 0.50 | 6 | 10-12 | 0.12 | 0.22 | 0.43 | 0.86 | 0.75 |
| 7 | 8-10 | 0.28 | 0.31 | 0.45 | 0.21 | 0.93 | 7 | 11-1 | -0.25 | 0.06 | 0.40 | 0.50 | 0.15 |
| 8 | 9-11 | 0.45 | 0.16 | 0.37 | 0.93 | 0.79 | 8 | 12-2 | 0.04 | 0.03 | 0.29 | 0.57 | 0.50 |
| 9 | 10-12 | 0.75 | 0.07 | 0.37 | 0.71 | 1.00 | 9 | 1-3 | -0.21 | 0.01 | 0.19 | -- | 0.36 |
| 10 | 11-1 | 0.12 | 0.10 | 0.42 | 0.57 | 0.25 | 10 | 2-4 | 0.22 | 0.14 | 0.27 | 0.36 | 0.56 |
| 11 | 12-2 | 0.12 | 0.02 | 0.30 | 0.57 | 0.70 | 11 | 3-5 | 0.18 | 0.22 | 0.35 | 0.55 | 0.57 |
| 12 | 1-3 | 0.14 | 0.03 | 0.19 | -- | 0.36 | 12 | 4-6 | 0.03 | 0.24 | 0.37 | 0.40 | 0.00 |

**Karnal 1**

| Month | Forecast Range | COR | BIAS[%] | RMSE[%] | AUC(-0.5) | AUC(0.5) | Month | Forecast Range | COR | BIAS[%] | RMSE[%] | AUC(-0.5) | AUC(0.5) |
|---|---|---|---|---|---|---|---|---|---|---|---|---|---|
| 1 | 2-4 | 0.42 | 0.02 | 0.48 | 0.50 | 0.71 | 1 | 5-7 | 0.24 | 0.06 | 0.31 | 0.70 | 0.57 |
| 2 | 3-5 | 0.05 | 0.06 | 0.47 | 0.22 | 0.67 | 2 | 6-8 | 0.66 | 0.17 | 0.27 | 0.78 | 1.00 |
| 3 | 4-6 | 0.49 | 0.00 | 0.38 | 0.50 | 0.86 | 3 | 7-9 | 0.50 | 0.10 | 0.17 | 0.80 | 0.61 |
| 4 | 5-7 | -0.30 | 0.09 | 0.35 | 0.45 | 0.14 | 4 | 8-10 | 0.32 | 0.12 | 0.19 | 0.64 | 0.50 |
| 5 | 6-8 | 0.45 | 0.16 | 0.28 | 0.83 | 0.63 | 5 | 9-11 | -0.16 | 0.00 | 0.31 | 0.56 | 0.21 |
| 6 | 7-9 | 0.58 | 0.10 | 0.17 | 0.80 | 0.94 | 6 | 10-12 | 0.11 | 0.48 | 0.66 | 0.50 | 0.57 |
| 7 | 8-10 | 0.53 | 0.17 | 0.21 | 0.71 | 0.63 | 7 | 11-1 | 0.01 | 0.51 | 0.60 | 0.39 | -- |
| 8 | 9-11 | -0.13 | 0.12 | 0.30 | 0.50 | 0.43 | 8 | 12-2 | -0.73 | 0.19 | 0.71 | 0.00 | 0.17 |
| 9 | 10-12 | 0.23 | 0.49 | 0.63 | 0.50 | 0.79 | 9 | 1-3 | 0.35 | 0.01 | 0.55 | 0.40 | 0.11 |
| 10 | 11-1 | 0.33 | 0.46 | 0.56 | 0.50 | -- | 10 | 2-4 | 0.00 | 0.03 | 0.54 | 0.70 | 0.44 |
| 11 | 12-2 | 0.40 | 0.13 | 0.56 | 0.80 | 0.50 | 11 | 3-5 | -0.36 | 0.17 | 0.56 | 0.50 | 0.50 |
| 12 | 1-3 | 0.45 | 0.02 | 0.52 | 0.80 | 0.61 | 12 | 4-6 | -0.15 | 0.03 | 0.41 | 0.29 | 0.43 |

**Arun**

| Month | Forecast Range | COR | BIAS[%] | RMSE[%] | AUC(-0.5) | AUC(0.5) | Month | Forecast Range | COR | BIAS[%] | RMSE[%] | AUC(-0.5) | AUC(0.5) |
|---|---|---|---|---|---|---|---|---|---|---|---|---|---|
| 1 | 2-4 | 0.38 | 0.09 | 0.30 | 0.71 | 0.56 | 1 | 5-7 | 0.24 | 0.01 | 0.14 | 0.61 | 0.57 |
| 2 | 3-5 | 0.13 | 0.20 | 0.30 | -- | 0.65 | 2 | 6-8 | -0.03 | 0.11 | 0.14 | 0.55 | 0.25 |
| 3 | 4-6 | 0.65 | 0.00 | 0.19 | 1.00 | 0.78 | 3 | 7-9 | -0.33 | 0.00 | 0.13 | 0.29 | 0.33 |
| 4 | 5-7 | 0.19 | 0.05 | 0.15 | 0.56 | 0.57 | 4 | 8-10 | 0.37 | 0.06 | 0.10 | 0.50 | 0.71 |
| 5 | 6-8 | 0.23 | 0.07 | 0.12 | 0.55 | 0.38 | 5 | 9-11 | 0.04 | 0.23 | 0.32 | 0.25 | 0.25 |
| 6 | 7-9 | 0.50 | 0.01 | 0.11 | 0.57 | 0.44 | 6 | 10-12 | 0.29 | 0.16 | 0.64 | 0.45 | 0.36 |
| 7 | 8-10 | 0.70 | 0.02 | 0.07 | 1.00 | 0.86 | 7 | 11-1 | 0.36 | 0.74 | 0.77 | 0.44 | -- |
| 8 | 9-11 | 0.29 | 0.17 | 0.26 | 0.75 | 0.75 | 8 | 12-2 | 0.09 | 0.44 | 0.68 | 0.35 | 0.63 |
| 9 | 10-12 | 0.05 | 0.24 | 0.57 | 0.50 | 0.36 | 9 | 1-3 | 0.40 | 0.10 | 0.51 | 1.00 | 0.79 |
| 10 | 11-1 | 0.21 | 0.73 | 0.76 | 0.50 | -- | 10 | 2-4 | 0.53 | 0.28 | 0.44 | 0.64 | 0.90 |
| 11 | 12-2 | 0.33 | 0.27 | 0.67 | 0.55 | 0.75 | 11 | 3-5 | 0.40 | 0.30 | 0.37 | -- | 0.78 |
| 12 | 1-3 | 0.19 | 0.12 | 0.60 | 0.21 | 0.43 | 12 | 4-6 | 0.03 | 0.05 | 0.23 | 0.39 | 0.33 |

*Tab. 2: Performance Measures for the F[1:3] and the F[4:6] model for all selected target areas. In order to better visualize the results, correlation and AUC values are colored using a color gradient from dark green (correlations and AUC near +1) and darkred (correlations near -1, AUC near 0). If no AUC values is provided, no drought/moist event has been observed during the evaluation period.*