# Peer review of "A statistically based seasonal precipitation forecast model with"

_Hydrology and Earth System Sciences, 2016_

## Referee Comment (RC1) · Anonymous Referee #1 · 8 Apr 2016

I only have some training in statistics, and don't have education background in hydrology or climate. My viewpoint may be quite different from the HESS community.

Technically, the forecast model is well-designed using various domain knowledge. I would like to know how kmeans is carried out, e.g., whether the correlation is used in the clustering, how to determine the random seeds or initial cluster centers. As we know, different random seeds will lead to different clustering results in K-means. Simply aggregating neighboring grids to a big region does not make sense if these grids have quite different correlation with the precipitation in the target region.

How can simply aggregating 'poor' monthly forecast to seasonal forecast lead to some good forecast results? Before using random forest, we should explore a single regression tree for monthly forecast first in order to identify which kinds of predictors are more important as well as its performance.

Reasonable forecast comparison should be carried out and reported. For example, what is difference if only those widely accepted climate indexes are used as predictors. How about comparing random forecast with a single regression tree. How about using 7 clusters instead of 7?

The sensitive analysis to me does not fit the key statement of paper that much. Actually from the appearance frequency of predictors in random forest can give us some ideas how important is a predictor.

Line 42, Page 15 has some typo or overclaim. AUC values are not always >0.7.

There are too many references. Reference Chen et al (2012) needs some correction.
* * *

---

## Referee Comment (RC2) · M. Barlow (Referee) · 20 May 2016

This is a very interesting approach and I enjoyed reading the paper. I do have some questions relating to data quality, statistical significance if the method is automated to a large number of regions, the very high forecast correlations apparently obtained, and the SST predictor regions for the Central Asia locations. Based on these questions, my recommendation is for major revisions. I do not have sufficient expertise to provide any detailed comments on the technical aspects of the cell-forest forecasting methodology.

Major comments:

1. Data quality. If the methodology is applied in an automated way to a number of different regions, how can data quality issues, which can vary considerably from location to location, be dealt with? And specifically for the case of Central Asia, the number of reporting stations varies dramatically over the 1948-2014 period considered here. I think the authors need to comment on both the general issue and provide some more information for the specific cases of Central and South Asia (e.g., plot the number of reporting stations as a function of time and assess the sensitivity of their results to the number of stations).

2. False positives if automated. Additionally, if the method is run for a large number of locations, some regions will get high prediction skill purely by chance. (If, say, a 95% significance criterion is applied for the validation period for each location, approximately 5% of the locations will appear significant by chance.) How would this issue be dealt with?

3. Forecast correlation magnitude. I'm somewhat confused by Table 2. Are the correlations for the training period or for the evaluation period? And is the seasonal cycle included when calculating the correlation or is it removed first? If not removed, then numbers for when it has been removed should also be shown. If I'm reading the table correctly, there are several forecast correlations between 0.7 and 0.86 – I'm not aware of any forecast correlations for precipitation (with seasonal cycle removed) that are anywhere near that high for any region using any forecast method. As an example, it appears that the forecast correlation for Naryn is 0.86 for JFM forecast from Dec. As far as I know, that's also considerably higher than any potential predictor for the region (SSTs, lagged precipitation, etc.). If I've read that correctly, that's a rather extraordinary result that will require extra evidence to be considered plausible – perhaps by identifying a few individual high-correlation predictors and showing that they are linearly independent. It would also be useful to put those numbers into the context of other reported forecast skill for the regions, especially from the usual seasonal forecast centers, and of the skill of a pure persistence forecast.

4. SST relationship for Central Asia. For the March SST correlations shown in Fig. 2, I don't understand why there is no signal at the equator in the central Pacific – I was expecting an ENSO pattern (and that is also what I get if I do a quick correlation based on GPCP data).

Minor comments:

1. I found the use of "exemplarily" to be somewhat distracting. I would suggest something more like "the model was applied to two test cases" or "two example cases." If the two regions really are exemplars, what makes them particularly useful or representative of the approach? Were other regions considered and, if so, why were they not included?

---

## Author Comment (AC1) · 3 Jun 2016

Dear referee,

thank you very much for your comments concerning our manuscript and particularly for the detailed remarks on the utilized statistical techniques.

Please find enclosed our response as well as some suggestions which will hopefully improve the presented manuscript.

Best regards, Lars Gerlitz et al.

[Figure]

###############################################################

1) I only have some training in statistics, and don't have education background in hydrology or climate. My viewpoint may be quite different from the HESS community. Technically, the forecast model is well-designed using various domain knowledge. I would like to know how k-means is carried out, e.g., whether the correlation is used in the clustering, how to determine the random seeds or initial cluster centers. As we know, different random seeds will lead to different clustering results in K-means.

The cluster analysis is performed using the Algorithm of Hartigan & Wong (1979), which is generally known as reliable and computationally efficient. The seeds are set randomly and are recursively updated afterwards in order to minimize the sum of squares between the observations and their assigned cluster center. In theory, the approach might be slightly sensitive to the choice of seeds, however, in practice we did not experience any variations of the clustering solution. In the revised manuscript, we will give some advanced information concerning the clustering techniques.

Hartigan, J. A. and M. A. Wong (1979) : A k-means clustering algorithm". Applied Statistics28.1, pp. 100-108.

2) Simply aggregating neighboring grids to a big region does not make sense if these grids have quite different correlation with the precipitation in the target region.

As stated in our manuscript, the clusters are not constructed based on spatial distances. The clustering routine is applied to the normalized time series of each potential predictor grid cell. This approach identifies clusters, which are characterized by a similar temporal variability of the considered predictor variable. Thus it is secured, that all grid cells within one cluster have a similar correlation with the precipitation time series.

3) How can simply aggregating 'poor' monthly forecast to seasonal forecast lead to some good forecast results?

Monthly precipitation amounts in general are characterized by a large (random) noise,

which is due to non-predictable meso- or local scale circulation patterns. The random noise often results in large magnitudes of variability on shorter time scales. On a seasonal scale, random events are most likely averaged out and the observed precipitation time series is less influenced by single events. Thus the hindcast based on large scale predictors can better reproduce the observations. We will clearly point that out in the revised manuscript.

4) Before using random forest, we should explore a single regression tree for monthly forecast first in order to identify which kinds of predictors are more important as well as its performance.

Due to the complex structure of the model and the high number of (partially highly correlated) predictor variables, we believe, that the investigation of single regression trees does not lead to a better interpretability of the model results. Single regression trees could only be constructed for one particular month and lead time, which would result in an overall number of 12*6=72 regression trees. The fact, that many of the predictor variables are correlated (e.g. the ENSO-variables and all of its covariates), impedes the interpretation of single regression trees.

5) Reasonable forecast comparison should be carried out and reported. For example, what is difference if only those widely accepted climate indexes are used as predictors. How about comparing random forecast with a single regression tree. How about using 7 clusters instead of 7?

The manuscript proposes one forecast methodology which especially stands out due to the negligence of traditional climate indices and the automatic predictor selection. Our main aim is to show, that a forecast based on data mining techniques without any prior knowledge is feasible. This has been shown by means of four case studies. We believe, that a comparison with different kinds of statistical models (traditional vs. automatic predictors / random forest vs. regression trees) would go beyond the scope of the manuscript.
Concerning the number of clusters, we absolutely agree, that this is a very subjective decision (which however might be a useful adjustment screw). As stated in the manuscript, "an excessive number of clusters might result in a disjunction of predictor regions, which reduces the predictive skill. On the contrary an insufficient number of clusters will lead to an aggregation of large regions which might still be characterized by a large inhomogeneity and thus are not suitable for the derivation of potential predictor variables" (p7l12). Results (not shown) indicate, that a low number of clusters leads to an underestimation of variability, while a large number of clusters rather leads to random like results. For future activities we will definitely think about automated solutions and will discuss this issue in the "discussion" part of the presented manuscript. Further we suggest to highlight the model sensitivity to the number of clusters in the "methods" section but resign from a comparison of different model results with variable cluster numbers. The latter would result in a confusing manuscript structure and would not create additional information.

6) The sensitive analysis to me does not fit the key statement of paper that much. Actually from the appearance frequency of predictors in random forest can give us some ideas how important is a predictor.

Unfortunately the calculation of simple predictor importance measures is only reliable for uncorrelated predictor variables. In general the random forest importance parameter for one particular predictor variable is based on the increase of the model error, which results from the modification of this variable (permutation importance). If there are highly correlated variables in the predictor space, every variable can easily be substituted, which results in unrealistically low values of the random forest importance measure

see e.g.: Gregurutti et al., 2014,: Correlation and variable importance in random forests, Statistics and Computing, DOI: 10.1007/s11222-016-9646-1).

Furthermore the aggregation to seasonal precipitation forecasts leads to a black boxlike model structure and impedes the interpretability. Thus we believe, that the sensitivity / response analysis based on traditional climate indexes is a good possibility to test the plausibility of the model. Since the results agree with former studies on the inter-annual precipitation variability in Central and South Asia, we argue that the sensitivity analysis gives evidence, that the model is able to find important variables (among others, those, which represent well known atmospheric modes) and to quantify their contribution. In a revised version we will try to better communicate the reasons for and the results of the sensitivity analysis.

7) Line 42, Page 15 has some typo or overclaim. AUC values are not always >0.7.

Will be changed.

8) There are too many references. Reference Chen et al (2012) needs some correction.

We will try to shorten the introductory section in some parts and correct the mentioned reference!

---

## Author Comment (AC2) · 3 Jun 2016

Dear Mr. Barlow,

thank you very much for your comments on our manuscript. Especially your remarks on the quality of gridded precipitation data sets and the model evaluation are highly appreciated.

Please find enclosed our response as well as some suggestions for improvement.

###############################################################

[Figure]

Major comments: 1. Data quality. If the methodology is applied in an automated way to a number of different regions, how can data quality issues, which can vary considerably from location to location, be dealt with? And specifically for the case of Central Asia, the number of reporting stations varies dramatically over the 1948-2014 period considered here. I think the authors need to comment on both the general issue and provide some more information for the specific cases of Central and South Asia (e.g., plot the number of reporting stations as a function of time and assess the sensitivity of their results to the number of stations).

We totally agree on the fact, that the quality of gridded precipitation often does not suit the demands of statistical analyses. Especially for Central Asia a distinct decrease of meteorological stations after 1990 has frequently been reported. In the revised manuscript we will better comment on the general data quality issues and provide some more information for the selected target areas. Further we suggest to include a homogeneity analysis in the model results, which can serve as a basis for the data quality assessment and the model interpretation.

2. False positives if automated. Additionally, if the method is run for a large number of locations, some regions will get high prediction skill purely by chance. (If, say, a 95% significance criterion is applied for the validation period for each location, approximately 5% of the locations will appear significant by chance.) How would this issue be dealt with?

In order to avoid the use of potential predictors, which are only correlated by chance, we splitted the predictor selection and the model calibration into two (independent) parts. "In order to avoid overfitting and to develop robust regression relationship, the model calibration is based on the second random sample and thus is independent from the predictor selection procedure" (p7l33). The potential predictors are based on the correlation analysis and will certainly include false predictors. E.g. not all of the predictor clusters shown in Fig. 2 will have a real forecast potential. The final regression tree algorithm however only selects those variables, which also explain some part of

the variance of the independent second sample. Thus the two step predictor selection procedure should deal with that problem.

3. Forecast correlation magnitude. I'm somewhat confused by Table 2. Are the correlations for the training period or for the evaluation period? And is the seasonal cycle included when calculating the correlation or is it removed first? If not removed, then numbers for when it has been removed should also be shown. If I'm reading the table correctly, there are several forecast correlations between 0.7 and 0.86 – I'm not aware of any forecast correlations for precipitation (with seasonal cycle removed) that are anywhere near that high for any region using any forecast method. As an example, it appears that the forecast correlation for Naryn is 0.86 for JFM forecast from Dec. As far as I know, that's also considerably higher than any potential predictor for the region (SSTs, lagged precipitation, etc.). If I've read that correctly, that's a rather extraordinary result that will require extra evidence to be considered plausible – perhaps by identifying a few individual high-correlation predictors and showing that they are linearly independent. It would also be useful to put those numbers into the context of other reported forecast skill for the regions, especially from the usual seasonal forecast centers, and of the skill of a pure persistence forecast.

The assessment of the forecast skill is an important task, which we already intensively discussed in our working group. The results which are shown in Tab. 2 are indeed calculated after removing the seasonal cycle. The correlation for DJF was e.g. calculated under consideration of those winter values only. However, as stated in the manuscript, the results should be interpreted with care, since the validation is only conducted for the independent time series, which includes 10 years of observations. "Although, the analysis based on 10 years only might be insufficient for the precise quantification of statistical model skill, we assume that a general assessment of the model quality is feasible. We abstained from the implementation of a cross-validation procedure due to the high computational demands of the predictor selection routine" (p9l17). Particularly for Naryn, the precipitation variability during the evaluation decade is strongly

linked to ENSO. Thus, the correlations might be distinctly lower in other decades, when the variability of ENSO is less pronounced.

We suggest to better comment on those issues in the manuscript. Further, we could conduct a split sample test, i.e. apply the predictor selection, model calibration and evaluation to shifted time periods. E.g. a second model evaluation based on the period from 1995 to 2004 would be feasible.

A disadvantage might be that this procedure would lead to a second large table and a duplication of Fig. 6, which might be slightly confusing for the reader. So we would be grateful for a final recommendation concerning the model validation strategy,

4. SST relationship for Central Asia. For the March SST correlations shown in Fig. 2, I don't understand why there is no signal at the equator in the central Pacific – I was expecting an ENSO pattern (and that is also what I get if I do a quick correlation based on GPCP data).

The attached figure shows the complete map of correlations between March precipitation time series in Naryn and the SSTs in previous January. Indeed, the entire ENSO region is positively correlated. However, highest levels of statistical significance are reached in those clusters which are highlighted by the polygons. Likewise the Warmpool region around Indonesia certainly reflects the ENSO signal. In order to avoid confusion, we will clearly comment on that point in the revised manuscript.

Minor comments: 1. I found the use of "exemplarily" to be somewhat distracting. I would suggest something more like "the model was applied to two test cases" or "two example cases." If the two regions really are exemplars, what makes them particularly useful or representative of the approach? Were other regions considered and, if so, why were they not included?

We will rather use the terms case studies or test cases in the revised manuscript. Although the study has been conducted in the framework of a research project dealing

with the forecast of hydro-climatological conditions in Central Asia, we aim at the development of a statistical model which can be easily transferred to other regions.

[Figure]

[Figure]

**Fig. 1.**

---

## Author Response (AR1)

**Dr. Lars Gerlitz**
**Deutsches GeoForschungszentrum - GFZ**
**Sektion 5.4 – Hydrologie**
**Haus C4**
**Telegrafenberg, 14473 Potsdam**
lars.gerlitz@gfz-potsdam.de
Telefon: +49 (0)331 288-28990

Potsdam, August 17th 2016

Dear Ladies and Gentleman,

please find enclosed our revised manuscript „A statistically based seasonal precipitation forecast model with automatic predictor selection and its application for Central and South Asia".
Please note, that the Title of our manuscript has been slightly changed and that we modified the order of the contributing authors.

As recommended by our reviewer Matthew Barlow and by the Editor Mr. Wang, we completely revised our evaluation strategy in order to better quantify the overall skill of our precipitation forecast model. Therefor we conducted a 4-fold split sample test, which is describes in detail in the manuscript. The use of the same data set for predictor selection and model calibration (as proposed by Mr. Wang) is unfortunately not feasible since it leads to strong overfitting and thus to a considerable decline of the model skill.

Since it has been seen, that the model skill for the small headwater catchments is low, we now applied the model to three larger target regions in Central and South Asia. The increase of the size of the target regions resulted in a distinct improvement of the model results, indicating that precipitation amounts in small catchments are highly variable due to mesoscale atmospheric processes which are difficult to forecast.

Further, we revised our sensitivity study. We believe that the description of the approach is better comprehensible in the revised manuscript and think that this analysis gives some advanced insights into the large scale climate and precipitation variability of our target regions.
However, we resigned from the utilization of alternative approaches (such as Bayesian Model Averaging), but might consider them for our future model development.

Eventually, as we assured in our primary response, we tried to consider all minor comments of the reviewers.

We believe that the manuscript has been improved considerably and again would like to thank the editor and the reviewers for their recommendations.

Yours,

Lars Gerlitz et al.

Helmholtz-Zentrum Potsdam · Deutsches GeoForschungsZentrum GFZ · Stiftung des öffentlichen Rechts
Telegrafenberg · 14473 Potsdam · www.gfz-potsdam.de
Vorsitzender des Kuratoriums: MinDir Dr. Karl Eugen Huthmacher
Vorstand: Prof. Dr. Dr. h.c. Reinhard Hüttl (Sprecher) · Dr. Stefan Schwartze
Bankverbindung: Konto-Nr. 3093887 · Deutsche Bank · BLZ 120 700 00
IBAN DE86120700000309388700 · BIC Code DEUTDEBB160 · Steuernr: 046/149/01166 · VAT DE138407750

---

## Author Response (AR2)

Dear Mr. Wang,

Thank you very much for your detailed comments concerning our manuscript HESS 2016-84.
A final revised version has now been uploaded.

We corrected the typos and changed some wordings, as suggested in your sticky notes.
Further we tried to address your main comments as follows:

(1) We tried to better explain the forecast aggregation strategy. Further we agree that a fixed term for the aggregated forecast improves the comprehensibility of the manuscript. We chose the term composite forecast and used it thoughout the manuscript. In our opinion, the equations on p. 11 seem to be correct.

*The F[1:3] $^m$-forecast model is defined as the sum of random forest model results based on predictor variables from the month m with lead times of 1, 2 and 3 months. The F[4:6] $^m$-forecast is equally based on predictor variables from month m, but involves the random forest models with lead times of 4, 5 and 6 months.*

$$F[1:3]^m = \sum_{l=1}^{3} RF(m,l) \ \& \ F[1:3]^m = \sum_{l=4}^{6} RF(m,l)$$

*where RF(m,l) is the specific Random Forest forecast model based on predictor variables of the month m and precipitation anomalies occurring after a lead time of l months. As an example, the F[1:3]$^{12}$-composite forecast including January, February and March is defined as the sum of three RF model results, which are all based on predictor variables from previous December. RF(m=12,l=1) utilizes predictor variables from December for the January forecast, RF(m=12, l=2) indicates the December based forecast for February, RF(m=12,l=3) is the forecast march.*

(2) We agree, that our modelling approach is actually deterministic, although, we describe a simple method for a transformation to a probabilistic forecast in the model evaluation section. In the revised manuscript, we shortly discuss this drawback in the summary and outlook section and suggest the generation of a model ensemble and a bayesian model averaging approach (as proposed in your JClim paper) as one possible improvement.

We are looking forward for your response. With best regards,

Lars Gerlitz et al.

---

## Author Response (AR3)

HELMHOLTZ-ZENTRUM POTSDAM
**DEUTSCHES**
**GEOFORSCHUNGSZENTRUM**

**Dr. Lars Gerlitz**
**Deutsches GeoForschungszentrum - GFZ**
**Sektion 5.4 – Hydrologie**
**Haus C4**
**Telegrafenberg, 14473 Potsdam**
lars.gerlitz@gfz-potsdam.de
Telefon: +49 (0)331 288-28990

Potsdam, October 10th 2016

Dear Ladies and Gentleman,

We are very pleased that our manuscript "A statistically based seasonal precipitation forecast model with automatic predictor selection and its application to Central and South Asia" has been accepted for final publication.

Once again I would like to kindly ask you to change the order of authors as following:

*Lars Gerlitz, Sergiy Vorogushyn, Heiko Apel, Abror Gafurov, Katy Unger-Shayesteh, and Bruno Merz*

Best regards,

Lars Gerlitz et al.

Helmholtz-Zentrum Potsdam · Deutsches GeoForschungsZentrum GFZ · Stiftung des öffentlichen Rechts
Telegrafenberg · 14473 Potsdam · www.gfz-potsdam.de
Vorsitzender des Kuratoriums: MinDir Dr. Karl Eugen Huthmacher
Vorstand: Prof. Dr. Dr. h.c. Reinhard Hüttl (Sprecher) · Dr. Stefan Schwartze
Bankverbindung: Konto-Nr. 3093887 · Deutsche Bank · BLZ 120 700 00
IBAN DE86120700000309388700 · BIC Code DEUTDEBB160 · Steuernr: 046/149/01166 · VAT DE138407750